# Comparison of a Barcode-Based Smartphone Application to a Questionnaire to Assess the Use of Cleaning Products at Home and Their Association with Asthma Symptoms

**DOI:** 10.3390/ijerph18073366

**Published:** 2021-03-24

**Authors:** Pierre Lemire, Sofia Temam, Sarah Lyon-Caen, Catherine Quinot, Etienne Sévin, Sophie Remacle, Karine Supernant, Rémy Slama, Orianne Dumas, Valérie Siroux, Nicole Le Moual

**Affiliations:** 1Université Paris-Saclay, UVSQ, Université Paris-Sud, Inserm, Équipe d’Épidémiologie Respiratoire Intégrative, CESP, 94807 Villejuif, France; pierre.lemire@inserm.fr (P.L.); STEMAM@mgen.fr (S.T.); catherine.quinot@gmail.com (C.Q.); sorem974@gmail.com (S.R.); orianne.dumas@inserm.fr (O.D.); nicole.lemoual@inserm.fr (N.L.M.); 2MGEN Foundation for Public Health (FESP-MGEN), 75748 Paris, France; 3University Grenoble Alpes, Inserm, CNRS, Team of Environmental Epidemiology Applied to Reproduction and Respiratory Health, Institute for Advanced Biosciences (IAB), 38000 Grenoble, France; sarah.lyon-caen@univ-grenoble-alpes.fr (S.L.-C.); karine.supernant@univ-grenoble-alpes.fr (K.S.); remy.slama@univ-grenoble-alpes.fr (R.S.); 4Epiconcept, 75011 Paris, France; e.sevin@epiconcept.fr

**Keywords:** household cleaning products, asthma, smartphone application

## Abstract

Household disinfectant and cleaning products (HDCPs) assessment is challenging in epidemiological research. We hypothesized that a newly-developed smartphone application was more objective than questionnaires in assessing HDCPs. Therefore, we aimed to compare both methods, in terms of exposure assessments and respiratory health effects estimates. The women of the SEPAGES birth cohort completed repeated validated questionnaires on HDCPs and respiratory health and used an application to report HDCPs and scan products barcodes, subsequently linked with an ingredients database. Agreements between the two methods were assessed by Kappa coefficients. Logistic regression models estimated associations of HDCP with asthma symptom score. The 101 participants (18 with asthma symptom score ≥1) scanned 617 different products (580 with available ingredients list). Slight to fair agreements for sprays, bleach and scented HDCP were observed (Kappa: 0.35, 0.25, 0.11, respectively). Strength of the associations between HDCP and asthma symptom score varied between both methods but all odds ratios (OR) were greater than one. The number of scanned products used weekly was significantly associated with the asthma symptom score (adjusted-OR [CI 95%]: 1.15 [1.00–1.32]). This study shows the importance of using novel tools in epidemiological research to objectively assess HDCP and therefore reduce exposure measurement errors.

## 1. Introduction

The deleterious effect of disinfectants and cleaning products (DCPs) on asthma has been well documented, especially in professional settings, but less is known regarding household exposures of which assessment is challenging. The use of household DCPs (HDCPs) is common in France, and in 2009 almost 80% the population considered that their use is not, or minimally, a risk for their health [1]. Accordingly, when buying a cleaning product, the main concerns for the customers are the price and the effectiveness of the products, with low interest for safety [2]. In a recent review, HDCPs were suggested as contributors to poor indoor air quality [3], especially for organic volatile compounds (e.g., limonene) and carbonyls (e.g., formaldehyde) which have known respiratory health effects. Noteworthy, in a Spanish case study [4], a higher frequency and intensity of HDCP use was observed during COVID-19 lockdowns, which may contribute to higher indoor air pollution. In addition, there is epidemiological evidence for adverse respiratory health effects of HDCPs [5], especially for those in spray form [6,7,8]. Nonetheless, specific chemical ingredients at risk are poorly identified and further research is needed.

In epidemiologic studies, household cleaning is classically self-reported through standardized questionnaires. However, self-reported exposure assessment may be prone to measurement biases, potentially differential [9,10]. In addition, specific HDCP ingredients are mostly unknown to customers, which could lead to under-reporting or misreporting of exposures [10]. More objective methods are warranted to improve the exposure assessment reliability, to handle the diversity of the products used and to study not only groups of products, but specific ingredients [11].

Exposure assessment by a BarCode-based smartphone Application (BC-App) is considered to be less prone to differential misclassification biases, more reliable, and to be less costly and time consuming than paper-based questionnaires [12]. Barcodes, in addition to the name and brand of the product, may also allow recording of the precise list of ingredients for each product [11,13]. A BC-App has been used in a professional setting to evaluate DCP exposure [11] among 14 workers from a French hospital. Participants scanned the barcode of DCPs used weekly at work and answered a short exposure questionnaire for each recorded product. DCP information mainly came from an existing database (http://www.prodhybase.fr/; accessed on 25 February 2021), which includes commercialized products used in hospitals, as registered by the manufacturer. Products information database was completed when necessary by the additional ingredients of products specifically used in the studied hospital. Recently, a similar BC-App to record HDCPs was used by volunteers of the ongoing SEPAGES birth cohort [14].

We aimed to compare two methods to evaluate HDCP exposure: a BC-App and a questionnaire. We also studied how exposures evaluated through both methods were associated with respiratory symptoms among women.

## 2. Materials and Methods

The SEPAGES (https://sepages.inserm.fr/; accessed on 25 February 2021) couple-child cohort aimed to study the effects of environmental exposures on pregnancy outcomes and child health [14]. Briefly, the cohort recruited 484 pregnant women within the first trimester of pregnancy between 2014 and 2017, but also their partner and future child. Pregnant women recruited in eight obstetrical ultrasonography practices located in the Grenoble area (France) had to fulfill the following eligibility criteria: being pregnant by less than 19 gestational weeks at inclusion, older than 18 years old, having a singleton pregnancy, planning to give birth in one of the four maternities clinics from Grenoble area and living in the study area. The women follow-up includes online and face-to-face questionnaires, clinical examinations, and environmental samples. Home visits for environmental sampling and respiratory health questionnaires interviews were scheduled at 4 data collection times: end of the first trimester (T1; 18 gestational weeks), end of the third trimester (T3; 34 gestational weeks), two months after delivery (environmental sampling only, M2) and one year after delivery (respiratory health only, Y1) (Appendix A). Starting in January 2017, in addition to online exposure questionnaires, participants were invited by a fieldworker to use for a week a BC-App to assess HDCP. Participants with exposure data assessed at least once by both methods and with complete health questionnaire data were included in the analysis (Figure 1).

HDCPs were evaluated by two methods: an online questionnaire (15% of the participants preferred to complete a postal questionnaire instead of the online one) and a BC-App. Each method was used a maximum of three times for each woman at the following three data collection times (Appendix A): T1, T3 and M2. Data recorded through the BC-App was secondarily linked to a HDCP database, referencing ingredients for each of the participants-scanned barcodes.

Participants reported their HDCP use over the last 3 months preceding the data collection times (T1, T3, M2; see Appendix A), using online standardized questionnaires. Frequency of weekly use of HDCPs was reported in four classes (never, less than once a week, 1–3 days per week, 4 to 7 days per week (daily use)), as done in previous surveys [6,15]. In the online questionnaires, frequency was recorded for a limited amount of HDCP categories (spray, irritants, bleach, scented products). Sprays, irritants and scented HDCP use was defined as the maximum frequency of use over the following categories: (a) sprays: furniture, floor, oven, glass, air fresheners and others (b) irritants: bleach, ammonia, solvents and acids (c) scented HDCP: scented HDCP, liquid or solid air fresheners, electric air fresheners and sprayed air fresheners. We defined bleach, spray, irritants and scented products frequency of use in a 2-class variable: non-users (“never” or “less than once per week”) and users (“1–3 days per week” or “4–7 days per week”). Furthermore, for sprays, irritants and scented HDCP we defined: (a) a 3-class variable: non-users (“never” or “less than once a week”), 1 product used weekly (one category with a frequency of use of “1–3 days per week” or “4–7 days per week”), 2 or more products used weekly (2 or more categories with a frequency of use of “1–3 days per week” or “4–7 days per week”); (b) a continuous variable: the sum of their defining HDCP categories used at least once a week.

The development of the application was done by Epiconcept (http://www.epiconcept.fr/en; accessed on 25 February 2021), a company specialized in the development of information systems for public health and the Voozanoo framework. The application used in the present study is an adapted version to the household cleaning context of an application previously used in a professional context [11].

Over the course of a week, volunteer participants used the BC-App, preinstalled on a provided phone, to scan the barcode and complete an in-application questionnaire for each HDCP commonly used by the household. For the analyses, we obtained four different types of data from the application for each product used: (a) barcodes (scanned); (b) frequency of weekly use (in four classes: “less than once”, “once”, “2–3 times” and “4–7 times”) (c) form (“spray”, “gel”, “liquid”, ”tablet”, “cream”, “foam”, ”crystals”, ”swipes”, ”wax”, ”diffuser”, other (free field)); (d) name (free field).

We used barcodes and brand names to record the most precise and up-to-date ingredients list of each product, freely available on the brand name website. Therefore, a prospective barcode-ingredients database of around 2000 HDCPs was built [16]. The database was updated by the HDCPs scanned by the SEPAGES volunteers. All products scanned by the volunteers have been checked/updated regarding ingredients and product form (Sophie Remacle). For each participant, the number of scanned HDCP used at least weekly was counted and two variables were defined: (a) a binary variable: “users” defined by weekly use of at least one product vs. “non-users”; (b) a 3-class variable defining the number of products used weekly: no product, one product, and at least two products.

For the present analyses, we selected three categories of ingredients: bleach, ammonia and scented HDCPs, which are the only ones that could be compared by both methods. Ingredients names are not standardized across exhaustive ingredients lists and Chemical Abstract Service (CAS) registry names are not required in the list of ingredients of a given cleaning product. A detailed list of ingredients names for each category of interest is available (Appendix A). Reported and registered names of products were also text-scanned for mentions of specific keywords to help products classification (Appendix A). We created 2-class and 3-class exposure variables for the number of scanned sprays, bleach and scented HDCPs, in addition to continuous variables defined previously. Moreover, we considered the mean number of ingredients of all HDCPs used weekly.

The asthma symptom score consists of the sum of positive answers to questions investigating five asthma-like symptoms reported in the last 12 months: (i) breathless while wheezing, (ii) woken up with a feeling of chest tightness, (iii) attack of shortness of breath at rest, (iv) attack of shortness of breath after exercise, (v) woken by attack of shortness of breath. The asthma symptom score has been described elsewhere [17], and is commonly used in the literature. In the analysis, the asthma symptom score (range: 0 to 5) is considered in two classes (0 vs. ≥1) as the number of participants who had a >1 asthma symptom score was low (T1: *n* = 6, T3: *n* = 4, Y1: *n* = 1). The asthma symptom score questionnaire was administrated by a fieldworker during home visits.

Age at baseline (first trimester of pregnancy) was considered as a continuous variable. Smoking status was defined as a 2-class variable, with “current smokers” (at least 1 cigarette per day between conception and pregnancy detection). Otherwise, participants were considered ‘non-current smokers”.

Comparison of HDCP evaluated through the BC-App and the online questionnaire was assessed by kappa coefficients, taking into account dependence of the kappas (due to repeated data) using a bootstrap method [18]. Hotelling’s T test was used to test differences between the 3 sub-kappas calculated at each time point. Agreement strength for kappa coefficients was interpreted as recommended [19]: poor: < 0; slight 0–0.2; fair: 0.2–0.4; moderate: 0.4–0.6; substantial: 0.6–0.8; and almost perfect 0.8–1.

Associations between HDCP and the 2-level asthma symptom score were evaluated by a Generalized Estimating Equations model (PROC GENMOD; SAS), to account for repeated data, and adjusted for age and smoking status.

Analyses were performed using SAS version 9.3 (SAS Institute Inc., Cary, NC, USA) and R version 4.0.2 and Rstudio interface version 1.3.1073.

## 3. Results

### 3.1. Description of the Study Population

Participation rates for the BC-App was similar to the online questionnaire for the first two data collection times (T1: 46/48 = 96% vs. 436/484 = 90%, T3: 89/98 = 89% vs. 383/484 = 79%, respectively), and lower for the last data collection time (M2: 83/146 = 57% vs. 370/484 = 76%, respectively). A total of 101 women had questionnaire data available for respiratory health (*n* = 286 observations) and for at least one of the studied products (bleach-based, sprayed and scented products; 291 observations), and had used the BC-App at least once out of the three collection times (197 observations) (Figure 1). Among the excluded participants, 58 were excluded due to missing health questionnaire or exposure data at the different follow-up and 325 had never used the BC-App (for most of them the BC-App has never been proposed at M2, especially for mothers included in 2015 in the study). Participants excluded for having never used the BC-App were not statistically different from the included population (Appendix A).

Participants were aged 32.5 years and 11% were current smokers between conception and pregnancy detection (Table 1). At inclusion 16% of the participants ever had asthma. The proportion of participants with at least one asthma symptom was 19%, 17%, and 6% at T1, T3 and M2, respectively. More than 80% of the women participated to the household cleaning tasks, and these tasks were shared with the partner for about 50% of the households.

Overall, 617 unique products (Table 2) over 197 week-long uses of the application were reported by the 101 women. HDCPs were diverse in their forms, with liquids being the majority (42%), followed by sprays (23%). A barcode and a list of ingredients were available for 94% of the 617 products (*n* = 580) and an ingredient list was reached for 554 products out of 580 (96%).

### 3.2. Comparison of the Two HDCP Assessment Methods

For each of the data collection times (T1, T3, M2) participants reported HDCP by both methods, retrospectively over the last 2 to 3 months for the online questionnaire and over the course of a week for the BC-App, over the same reporting periods (see Appendix A). From the application, 44%, 16% and 87% of women reported weekly use of spays, bleach and scented products. From the online questionnaire, prevalence of weekly use of sprays and bleach were similar (37% and 11%), but it was less than one-half (41%) for scented products. Concordance between both methods were around 70% for sprays, 80% for bleach and 50% for scented products, and did not vary between the three data collection times (Appendix A). The kappa coefficients (Table 3) showed slight agreement between records from the BC-App and reports from questionnaires for scented HDCP (0.11 [0.03–0.19]) and fair agreement for both bleach (0.25 [0.09–0.41]) and sprays (0.35 [0.18–0.51]). Kappa for ammonia could not be calculated as there is no users according to the online questionnaires. For each product, the kappa coefficient estimated at each data collection times (T1, T3, M2) did not differ.

### 3.3. Associations between Household Cleaning Products and the Asthma Symptom Score

A statistically significant positive association was observed between the total number of products used weekly (available only by the application) and the asthma symptom score, whereas no association was observed for the number of ingredients per product (Table 4). No statistically significant association between the use of spray, irritants, bleach and the asthma symptom score was observed, regardless of the method used to evaluate exposure data. However, adjusted OR estimates for both methods are nearly all >1 and we observed, for the online questionnaire, associations close to the borderline significant threshold (in bold) for scented HDCPs (*p* = 0.0572), irritants (*p* = 0.0679) and sprays (*p* = 0.0924). For scented HDCPs evaluated by the BC-App, only 24 participants reported not using any and we found a significant association between the number (continuous) of scented HDCPs used weekly and a positive asthma symptom score. Moreover, when defining weekly number of scented HDCPs in three classes (2 (*n* = 31), >3 (*n* = 85) versus ≤ 1 (*n* = 72)), risk of an asthma symptom score ≥1 increased gradually (OR = 2.05 [0.70–5.85] for 2 scented HDCP, OR = 2.71 [1.04–7.05] for at least three scented HDCPs, *p* for trend (*p* = 0.04)).

## 4. Discussion

This is the first study to compare two assessment methods of HDCP use: a newly developed BC-App and a standardized questionnaire. Our findings show slight to fair agreement between the weekly use of HDCPs assessed by a BC-App questionnaire as compared to an online questionnaire. When comparing health association results between both methods, the magnitude of the associations differed according to the method used, although most associations showed statistically non-significant OR greater than 1.

Comparison of agreement for HDCP use between the questionnaire and the BC-App is original, preventing any direct comparison with data in the literature. Agreement levels differences may partly be due to singularities of each category. The “spray” category is the only one which is directly observable by the participant. It would therefore be easier to memorize and classify for the participants and may explain the highest agreement level compared to scented HDCPs and bleach. However, products’ forms recorded in the ingredients database may have potential classification errors. Indeed, participants can repackage some liquid products (i.e., vinegar) in a spray form or create their own homemade product from raw products (i.e., essential oils and baking soda). Spray refills were categorized as liquids, and products without barcodes were accounted as missing data. In addition, participants may not have scanned air-refreshing sprays or plug-ins diffusers if they were not considered as HDCPs. Thus, assessment based on the BC-App linked with the ingredients database may under-estimate spray use in our study. The “scented products” had a slight agreement level, the lowest one. Based on the ingredients list to evaluate scented HDCPs, nearly all HDCPs contain perfumes. Participants may not be aware that their HDCPs are scented, which might explain the strong underestimation of scented HDCP use when using the questionnaire as compared with the application. Finally, “bleach” has a fair agreement: it is often stated in the commercial names of HDCPs that are composed of bleach and known to be a disinfectant. In another study, agreement level between self-reported and experts’ evaluation of bleach exposure was substantial, but under-estimated in the self-reports, especially among non-asthmatics [10]. Participants’ lack of knowledge about specific HDCP ingredients was hypothesized to have led to underreport of exposure by questionnaire as compared with expert assessment, except for sprays [10]. A similar bias may exist in our study and may explain the observed slight-to-fair agreement levels.

Using the questionnaire, associations with the asthma symptom score were consistent with previous studies for sprays [6,7,8,20], bleach [6] and number of irritants [15]. Taking into account the number of irritants used is recent, and the previous study on the French elderly E3N cohort [15] (women in average, 70 years old), observed positive associations between the number of irritant HDCPs and current asthma similar to our study. However, our results do not confirm previous positive associations observed for bleach [19] in the French EGEA asthma cohort, but this may be due to a generational effect (SEPAGES: mean age of 33 and 20% bleach use; EGEA: mean age of 45 and 40% bleach use). For scented HDCP, evaluated either by the online questionnaire or the application, the significant positive associations with the asthma symptom score in the present study, were stronger than associations with incident asthma (HR: 1.3 [0.7–2.3]) observed in the European Community Respiratory Health [6]. The authors of the later study hypothesized that those with asthma avoided such products, which may bias associations toward the null. This hypothesis is supported by a study comparing occupational HDCPs between a questionnaire and a job-exposure matrix [9], showing that asthma status influenced answers to the questionnaire. In our study, using the application, only 24 participants were evaluated as non-users of scented HDCPs, thus appropriate caution must be taken when interpreting the 2-class and 3-class variables. However, the continuous number of scented HDCPs used weekly is significantly associated with the asthma symptom score. Lastly, we investigated for the first time in the literature the association between the mean number of ingredients, contained in HDCPs used weekly, and asthma symptom score, but no significant association was observed. Ingredients data might be a key to study the mixture effect of exposure to several chemicals on respiratory health. Studies on larger population samples are needed to explore those opportunities. 

The main strengths of this study are the use for the first time in a population-based study of a recently developed BC-App to evaluate HDCPs and its comparison with a standardized questionnaire. The questionnaire used to evaluate HDCPs in the SEPAGES study has been previously used in domestic settings [6,7,8,15,21], among which significant associations between HDCPs and respiratory health were found. In addition, we linked barcode data from the application to their ingredients list for around 600 products. This two-step methodology was hypothesized to overcome questionnaire limitations, such as its reliance on participants’ knowledge about HDCP ingredients, a limited number of pre-defined products compounds and the fact that a unique product may be reported over several items in the questionnaire. Thus, the method based on the BC-App and the linked ingredient database was hypothesized to be more objective and to lead to less biased HDCP evaluations, especially for the number of products and ingredients contained in the products used weekly.

However, this study has some weaknesses. First, there was a limited amount of information that could be compared between the online questionnaire and the in-application questionnaires/linked database. Online questionnaires are, by construction and necessity, limited to rough categories of HDCPs, which included specific or non-specific chemicals (e.g., bleach, ammonia, acids), cleaning tasks (cleaning oven or floors), product forms (sprays) or combinations (degreasing sprays). Thus, one HDCP can appear in several questionnaire categories, and it can be hard to estimate the number of HDCPs a participant may use. The BC-App may circumvent these limits by having an objective list of ingredients, but this is also highly dependent on the quality of publicly available data on the brand website on products compounds, which are constantly evolving and sometimes hard to trace. Minimum European regulation [22] requirements of ingredients lists are also limited, as ingredients such as dyes and perfumes, outside of a limited list may not be stated precisely. For example, our results suggested associations between weekly use of scented HDCPs or perfumes consistently by both methods. However, it was not possible to evaluate the impact of the number of scented ingredients per product on the asthma symptom score, as, for most of the products, non-specific ‘perfume’ term (>90%; not shown) was indicated in the ingredients list instead or in addition to specific ones (limonene, etc.; see Appendix A). Rough weight ranges and Chemical Abstract Services (CAS) registry number for ingredients are also not mandatory and could be relevant in evaluating exposure in epidemiological studies. Due to the lack of ‘gold standard’, a validation of the method based on the in-application questionnaires and linked database is not possible. In addition, the assessment methods were not used strictly at the same times. The exposure questionnaire is available online for a longer period and retrospectively cover three months of HDCP use (Figure 1) whereas the BC-App is used over a single week. The online questionnaire may then cover several changes of habits, whereas by the application participants scanned products used during the week (and not all products owned), which could not reflect a usual week. A further limitation lies in the limited sample size, which impact the statistical power of the health effects estimates and prevent any sensitivity analyses. Although the asthma symptom score was recommended to be used as a continuous variable, it has been studied in a dichotomous way in our study due to the low prevalence of women with at least two asthma symptoms in the study population. In addition, we observed a variability in asthma symptoms prevalence during pregnancy, which may be partly explained by hormonal factors, that are likely involved in asthma expression and symptoms [23]. Nevertheless, we should emphasize here that the main objective of this study regards methodological aspects, more than the etiological aspects.

## 5. Conclusions

The newly developed BC-App to assess the use of DCPs at home in epidemiological research showed fair agreement for spray and bleach use and slight agreement for scented HDCPs when compared to a standardized questionnaire. Although associations with asthma symptoms showed similar directions with both assessment methods, magnitude of associations varied. Noteworthy, the application suggested a positive association between the number of products or scented HDCPs used weekly and symptoms of asthma. This study supports the use of a barcode-based application combined to an ingredients list database to identify HDCP ingredients as a promising tool in epidemiological research. Further research on a larger population is needed to confirm our findings and to identify specific ingredients from HDCPs, or mixture effects at risk for respiratory health among adults and children. In addition, while further studies are needed to identify reasons for the large discrepancy between the two assessment methods especially for scented products, our results suggest that the objective BC-App method may be more accurate and result in better assessment of associations with lung health outcomes.

## Figures and Tables

**Figure 1 ijerph-18-03366-f001:**
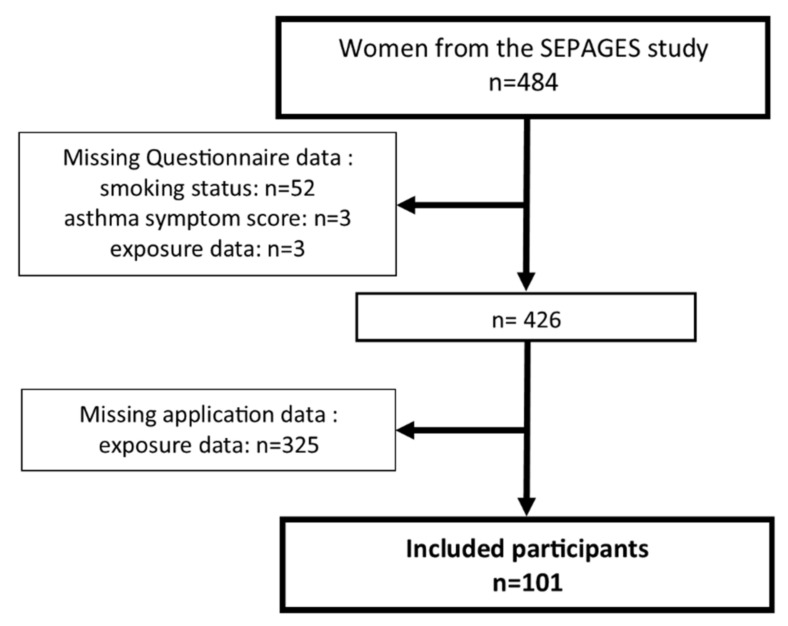
Flowchart for the selected population.

**Table 1 ijerph-18-03366-t001:** Population characteristics according to the three data collection times in the SEPAGES study.

	T1 ^d^	T3 ^d^	M2 ^d^
Questionnaire data	101		
** Age (years) ^a^, mean ± s.d**	32.5 ± 3.6		
** Smoking status ^b^**	101		
Current smoker	11 (10.9)		
** Asthma Symptoms Score ^c^**	96	95	95
≥1	18 (18.8)	16 (16.8)	6 (6.3)
** Household Help**	100	97	94
Participant alone	15 (15.0)	12 (12.4)	13 (13.8)
Participant and help	76 (76.0)	68 (70.1)	71 (75.5)
Help alone	9 (9.0)	17 (17.5)	10 (10.7)
** Weekly spray use**	91	92	92
Yes	34 (37.4)	35 (38.0)	33 (35.9)
** Nb of weekly spray use, among users**	34	35	33
1	22 (24.2)	24 (26.0)	19 (20.7)
≥2	12 (13.2)	11 (12.0)	14 (15.2)
** Weekly irritants use**	98	92	93
Yes	45 (45.9)	50 (54.3)	44 (47.3)
** Nb of weekly irritants use, among users**	45	50	44
1	39 (39.8)	38 (41.3)	38 (40.9)
≥2	6 (6.1)	12 (13.0)	6 (6.4)
** Weekly bleach use**	100	97	94
Yes	10 (10.0)	14 (14.4)	7 (7.5)
** Weekly scented products use**	98	94	92
Yes	42 (42.9)	39 (41.5)	33 (35.9)
** Nb of weekly scented products use, among users**	42	39	33
1	33 (33.7)	30 (31.9)	25 (27.2)
≥2	9 (9.2)	9 (9.6)	8 (8.7)
**Application data**	**42 ^e^**	**79 ^e^**	**76 ^e^**
** Weekly spray use**	42	79	76
Yes	21 (50.0)	34 (43.0)	33 (43.4)
** Nb of weekly spray use, among users**	21	34	33
1	10 (23.8)	19 (24.0)	19 (25.0)
≥2	11 (26.2)	15 (19.0)	14 (18.4)
**Application and ingredients data**	**42 ^e^**	**79 ^e^**	**76 ^e^**
** Number of products used weekly**	42	79	76
≥3	28 (66.7)	49 (62.0)	41 (54.0)
Median [Q1; Q3]	3 [2;5]	4 [2;6]	3 [1;5]
** Mean of ingredients used weekly**	42	79	76
Median [Q1; Q3]	11 [8;14]	11 [8;13]	11 [8;14]
** Weekly bleach use**	42	79	76
Yes	9 (21.4)	11 (13.9)	13 (17.1)
** Weekly scented products use**	42	79	76
Yes	40 (95.2)	67 (84.8)	66 (86.8)
** Nb of weekly scented products use, among users**	40	67	66
1	12 (28.5)	18 (22.8)	21 (27.6)
≥2	28 (66.7)	49 (62.0)	45 (59.2)

All data is in *n* (%), otherwise stated ^a^ before pregnancy: non-repeated data, ^b^ between conception and pregnancy detection: non-repeated data, ^c^ only data collected one year after delivery (Y1) instead of two months after delivery (M2),^d^ data collection times: first trimester of pregnancy(T1), third trimester of pregnancy(T3), second month after delivery(M2), ^e^ after exclusion, total of smartphone application users before exclusion: T1: 46, T3: 89, M2: 83.

**Table 2 ijerph-18-03366-t002:** HDCP characteristics among 101 women participating in the SEPAGES study for all smartphone datapoints.

Number of Application Uses (Unique Participants)	197(101)
** Number of unique products reported by the application ^a^**	617
**Linked Ingredients data**	
Missing ingredients, *n*(%)	37 (6.0)
Products format ^b^, *n*(%)	580
Liquids	242 (41.7)
Sprays	134 (23.1)
Gel	80 (13.8)
Tablets ^c^	34 (5.9)
Powder	32 (5.5)
Swipes	28 (4.8)
Others	30 (5.2)

^a^ each participant can have up to three distinct use of the application^, b^ from products database, corrected with format from participant when relevant for spray^, c^ includes “tablet”, “lozenge” and “block”.

**Table 3 ijerph-18-03366-t003:** Comparison of HDCPs assessed by questionnaire and smartphone application, all data collection points (*n* = 188).

Questionnaire Data	Smartphone Application Data	
	No	Yes
** Weekly spray use ^a^, *n***		
No	78	35
Yes	21	44
Kappa ^b^ coefficient [95% CI]	0.35 [0.18–0.51]	
Hotelling’s T Test ^c^	0.56	
** Weekly bleach use ^a^, *n***		
No	146	21
Yes	12	9
Kappa ^b^ [95% CI]	0.25 [0.09–0.41]	
Hotelling’s T Test ^c^	0.91	
** Weekly scented products use ^a^, *n***		
No	19	87
Yes	4	71
Kappa ^b^ [95% CI]	0.11 [0.03–0.19]	
Hotelling’s T Test ^c^	0.45	

^a^ Application data with linked ingredients data are compared to questionnaire data ^b^ measure of agreement between questionnaire and application data [19]: poor: <0; slight 0–0.2; fair: 0.2–0.4; moderate: 0.4–0.6; substantial: 0.6–0.8; and almost perfect 0.8–1, ^c^ test for difference between the 3 sub-kappas (one at each time of evaluation of HDCP).

**Table 4 ijerph-18-03366-t004:** Associations between household cleaning products use and ≥1 asthma symptom score, according to the methods of cleaning products use assessment.

	≥1 Asthma Symptom Score					
	Questionnaire Data ^a^			Smartphone and Ingredients Data ^a^		
	*n*	OR [95%CI]	OR ^b^ [95%CI]	*n*	OR [95%CI]	OR ^b^ [95%CI]
**Weekly products use ^c^, *n***	**Not applicable**			188		
Continuous				188	**1.15 [1.00–1.31]**	**1.15 [1.00–1.32]**
Number: 0–2 (ref)				75	1	1
≥3				113	**2.57 [0.99–6.71]**	**2.58 [0.98–6.78]**
**Weekly ingredients, mean**				188		
Continuous				188	1.02 [0.95–1.09]	1.02 [0.95–1.09]
**Weekly spray use ^d^, *n***	260			188		
Continuous	95	1.38 [0.87–2.17]	1.44 [0.90–2.32]	85	1.22 [0.84–1.77]	1.22 [0.84–1.77]
No (reference)	165	1	1	103	1	1
Yes	95	1.60 [0.83–3.09]	1.63 [0.85–3.15]	85	1.17 [0.53–2.59]	1.16 [0.47–2.57]
Number: 1 (vs. no)	61	1.18 [0.55–2.54]	1.15 [0.53–2.50]	50	1.07 [0.43–2.64]	1.06 [0.42–2.64]
≥2 (vs. no)	34	2.48 [0.81–7.61]	2.84 [0.90–8.99]	37	1.30 [0.48–3.57]	1.29 [0.47–3.54]
*p* for trend	260	0.1204	**0.0924**	188	0.5955	0.6100
**Weekly irritant use ^d^, *n***	268			**Not applicable**		
Continuous	268	1.51 [0.96–2.39]	1.50 [0.94–2.40]			
No (reference)	136	1	1			
Yes	132	1.86 [0.92–3.75]	1.83 [0.90–3.71]			
Number: 1 (vs. no)	109	1.70 [0.85–3.44]	1.68 [0.84–3.38]			
≥2 (vs. no)	23	2.61 [0.88–7.72]	2.58 [0.86–7.74]			
*p* for trend	268	**0.0583**	**0.0679**			
**Weekly bleach use ^d^, *n***	276			188		
No (reference)	245	1	1	157	1	1
Yes	31	1.18 [0.48–2.91]	1.13 [0.45–2.90]	31	1.06 [0.38–32.94]	1.04 [0.37–2.91]
**Weekly scented products use ^d^, *n***	269			188		
Continuous	269	**1.46 [0.99–2.14]**	**1.47 [1.00–2.17]**	164	**1.16 [1.02–1.32]**	**1.16 [1.02–1.32]**
No (reference)	162	1	1	24	1	1
Yes	107	1.69 [0.88–3.24]	1.74 [0.89–3.40]	164	6.20 [0.81–47.15]	6.21 [0.81–47.52]
Number: 1 (vs. no)	84	1.53 [0.75–3.13]	1.58 [0.77–3.24]	48	3.51 [0.34–36.53]	3.50 [0.34–36.15]
≥2 (vs. no)	23	2.38 [0.91–6.28]	2.51 [0.93–6.74]	116	7.87 [0.92–67.73]	7.95 [0.91–69.28]
*p* for trend	269	**0.0642**	**0.0572**	188	**0.0055**	**0.0063**

^a^ data source used for analyses; all collection times considered for each participant (repeated data analysis: generalized estimating equations model), ^b^ adjusted for age and smoking status (between conception and pregnancy detection), ^c^ total of cleaning products declared used of at least once a week, ^d^ reported use of at least once a week.

## Data Availability

Data were obtained from the SEPAGES study group. It can be provided upon reasonable request after approval by the SEPAGES steering committee.

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
