# Peer review of "Comparison of a Barcode-Based Smartphone Application to a Questionnaire to Assess the Use of Cleaning Products at Home and Their Association with Asthma Symptoms"

_ijerph, 2021, doi:10.3390/ijerph18073366_

Round 1

Reviewer 1 Report

Thank you for the opportunity to review this interesting and well-written paper. This is an important investigation as developing novel methods to more accurately assess exposure, especially in population based epidemiologic studies is very important. Reliance purely on questionnaire can lead to bias. The following are some comments and questions I have for the authors.

  1. This is from a large epidemiologic cohort that is referenced. However, there should be a brief description of the selection of the population [e.g. where was the study located, from where were the women recruited (hospital, clinic, etc.), were all eligible women approached, etc.].
  2. It is a little unclear to me who used/entered the information onto the app. Was it the study participants themselves? The text says volunteers, which could mean volunteer researchers. However, I interpreted the paper to suggest that participants used the app.
  3. Fifteen percent of the population completed postal questionnaires. What was the reason for this and were there differences in the responses to the online questionnaire group?
  4. Three categories of ingredients were considered. Why were only these three categories considered?
  5. The potential confounders considered were very limited. There are some important ones that should also be considered in this study including things such as presence of pets, mould, presence of children already, etc as these may be related to both specific cleaning product use (type and strength, etc) and asthma.
  6. Kappa values can be limited by prevalence resulting in reduced levels of agreement. Table 3 shows the contingency tables but it may be helpful to present %concondance and discordance as well. It may also be helpful to consider a Spearman’s correlation analysis.
  7. The authors discuss some of the reasons around the very large difference in use of scented products. While there needs to be some work done to further identify reasons for the large discrepancy, it could suggest that the objective methods may be more accurate and result in better assessment of associations with lung health outcomes.

Reviewer 2 Report

  1. Authors "hypothesize" that smartphone app is "more objective" than questionnaires, but they never answer this question. They acknowledge no gold standard but give little guidance on which method to use, or which is "more objective" or better.
  2. Did authors ask participants about history of asthma? It should be added if they did.
  3. The authors used just 1 asthma symptom to indicate  possible/probable asthma. Is this acceptable or useful, given the use of this asthma score in the past?
  4. Authors should explain why asthma score > 1 dropped so much in M2.
  5. Tables need re-formatting, esp. Table 1 and 4. Hard to read in current format.
  6. Table 3 shows discrepant reporting of sprays , bleaches, and scents was far greater on the app compared to the questionnaire. Authors should comment/explain.
  7. Results, Section 3.2: Did the app and the questionnaire cover the same reporting period?  This may be answered somewhere else in the manuscript, but, if so, should be repeated here.
  8. Table 4: I would bold the OR's with lower CI of 0.99, indicating borderline significance. But the authors overstate "borderline" in Section 3.3, line 7, since most of the CI's they refer to are not borderline in significance.
  9. Discussion. Paragraph #2, first sentence repeats idea of paragraph #1.
  10. See attached file for word suggested corrections.
